# Highly Sensitive Nanomagnetic Quantification of Extracellular Vesicles by Immunochromatographic Strips: A Tool for Liquid Biopsy

**DOI:** 10.3390/nano12091579

**Published:** 2022-05-06

**Authors:** Vera A. Bragina, Elena Khomyakova, Alexey V. Orlov, Sergey L. Znoyko, Elizaveta N. Mochalova, Liliia Paniushkina, Victoria O. Shender, Thalia Erbes, Evgeniy G. Evtushenko, Dmitry V. Bagrov, Victoria N. Lavrenova, Irina Nazarenko, Petr I. Nikitin

**Affiliations:** 1Prokhorov General Physics Institute of the Russian Academy of Sciences, 38 Vavilov St., 119991 Moscow, Russia; bragina_vera@nsc.gpi.ru (V.A.B.); elena.khomyakova@exosome-analytics.com (E.K.); alexey.orlov@kapella.gpi.ru (A.V.O.); znoykos@yandex.ru (S.L.Z.); mochalova@phystech.edu (E.N.M.); 2Moscow Institute of Physics and Technology, 9 Institutskii per., 141700 Dolgoprudny, Russia; 3Sirius University of Science and Technology, 1 Olympic Ave., 354340 Sochi, Russia; 4Institute for Infection Prevention and Hospital Epidemiology, Medical Center—University of Freiburg, Faculty of Medicine, University of Freiburg, 79106 Freiburg, Germany; paniushkina.liliia@gmail.com (L.P.); irina.nazarenko@uniklinik-freiburg.de (I.N.); 5Federal Research and Clinical Center of Physical-Chemical Medicine of the Federal Medical and Biological Agency, 1a Malaya Pirogovskaya St., 119992 Moscow, Russia; victoria.shender@gmail.com (V.O.S.); pkviktoria@mail.ru (V.N.L.); 6Shemyakin-Ovchinnikov Institute of Bioorganic Chemistry of the Russian Academy of Sciences, 16/10 Miklukho-Maklaya St., 117997 Moscow, Russia; 7Department of Obstetrics and Gynecology, Medical Center—University of Freiburg, Faculty of Medicine, University of Freiburg, 79106 Freiburg, Germany; thalia.erbes@uniklinik-freiburg.de; 8Lomonosov Moscow State University, 1 Leninskie Gory, 119991 Moscow, Russia; evtushenko@enzyme.chem.msu.ru (E.G.E.); bagrov@mail.bio.msu.ru (D.V.B.); 9German Cancer Consortium (DKTK), Partner Site Freiburg and German Cancer Research Center (DKFZ), 69120 Heidelberg, Germany; 10National Research Nuclear University MEPhI (Moscow Engineering Physics Institute), 31 Kashirskoe Shosse, 115409 Moscow, Russia

**Keywords:** antibody-functionalized magnetic nanoparticles, magnetic particle quantification, nonlinear magnetization, immunochromatographic test strips, extracellular vesicles, breast and ovarian cancers

## Abstract

Extracellular vesicles (EVs) are promising agents for liquid biopsy—a non-invasive approach for the diagnosis of cancer and evaluation of therapy response. However, EV potential is limited by the lack of sufficiently sensitive, time-, and cost-efficient methods for their registration. This research aimed at developing a highly sensitive and easy-to-use immunochromatographic tool based on magnetic nanoparticles for EV quantification. The tool is demonstrated by detection of EVs isolated from cell culture supernatants and various body fluids using characteristic biomarkers, CD9 and CD81, and a tumor-associated marker—epithelial cell adhesion molecules. The detection limit of 3.7 × 10^5^ EV/µL is one to two orders better than the most sensitive traditional lateral flow system and commercial ELISA kits. The detection specificity is ensured by an isotype control line on the test strip. The tool’s advantages are due to the spatial quantification of EV-bound magnetic nanolabels within the strip volume by an original electronic technique. The inexpensive tool, promising for liquid biopsy in daily clinical routines, can be extended to other relevant biomarkers.

## 1. Introduction

Extracellular vesicles (EVs) are promising agents for liquid biopsy—a non-invasive approach for diagnosis of cancer and evaluation of therapy response [1,2,3]. EVs are present in various body fluids, play a significant role in the regulation of tumor growth and metastasis [4,5,6,7], and may serve as a source of disease biomarkers [8,9]. The total EV population can be measured using their common membrane surface proteins, namely, CD9, CD63, and CD81 tetraspanins, even more accurately than with non-specific methods, such as the traditional nanoparticles tracking analysis (NTA) or protein quantification assays (e.g., bicinchoninic acid-based assay) [10,11]. Alongside the general membrane markers, EVs carry donor-cell specific proteins, e.g., epithelial cell adhesion molecules (EpCAMs) that are overexpressed in some solid tumors and extensively investigated for active targeting by nanodrugs [12,13]. Their quantification may allow estimation of the fraction of EVs specific to a particular pathology.

Despite the substantial progress in EV knowledge, only two major conventional technologies, namely, flow cytometry and enzyme-linked immunosorbent assays (ELISAs), have been widely used for EV quantification and characterization based on analysis of their membrane proteins [14,15,16,17,18,19]. Recent developments include EV detection by nuclear magnetic resonance [20], surface plasmon resonance (SPR) [21,22], Raman spectroscopy [23], surface-enhanced Raman spectroscopy (SERS) [21,24,25], fluorescent NTA [26], imaging flow cytometry (IFC) [27,28], single EV flow cytometry [29], and interferometry-based [30] methods. These techniques are highly sensitive but quite expensive, complicated, require highly qualified personnel, and are hardly applicable for routine medical diagnostics.

Easy-to-use lateral flow (or immunochromatographic) assays (LF or IC, respectively), widely used for rapid protein quantification [31,32], even at a point of care [33,34], are especially attractive for EV detection as a liquid biopsy tool. Recently, quantitative LF assays were proposed that offered high sensitivity and a wide dynamic range [35,36,37,38,39]. A recent study [40] reported LF systems that used membrane surface proteins as targets for EV determination with an optical reflectance reader using gold, carbon, and magnetic labels. The best limit of detection (LOD) of 3.4 × 10^6^ EVs/μL was achieved with gold nanolabels. However, the applied optical interrogation may limit the assay sensitivity and cannot reveal the power of magnetic nanolabels. Another type of registration of magnetic labels on IC strips, namely, impedance measurements, featured a LOD of 1 × 10^7^ EVs/μL [41].

A fundamentally different registration principle of magnetic particle quantification (MPQ) [42] has enabled development of the nanomagnetic IC tool presented here that quantitatively determines EVs isolated from cell culture supernatants and various body fluids. The MPQ principle employs a nonlinear magnetization of magnetic nanoparticles (MP) subjected to an alternating magnetic field at two frequencies to register particle response at a combinatorial frequency. This way, the signal is generated solely by the non-linear magnetic materials, while dia- and paramagnetics, such as biological entities, plastic, etc., do not contribute to the signal. The MPQ detectors permit registration down to 39 pg (or 87 pieces) of magnetic nanoparticles in a relatively large 0.2 mL volume with the record linear dynamic range of seven orders [42]. The MPQ registration technique has proved to be effective for highly sensitive detection of various agents in complex mediums (food, human serum, urine, etc.) [39,43], for comprehensive characterization of MP bioconjugate–target interactions [44], and various in vivo studies [45,46,47].

The inexpensive and easy-to-use nanomagnetic IC tool proposed here has a two orders better sensitivity (3.7 × 10^5^ EVs/µL) than those of the most sensitive traditional LF systems and commercial ELISA kits. We successfully tested the tool for quantification of EVs harboring common EV markers, namely, CD9 and CD81, and a tumor-associated biomarker EpCAM. The tool performance was verified by EV detection in clinical samples from patients with breast and ovarian cancers. An isotype control line, for the first time introduced to the IC strips, ensures the detection specificity. The methodological advantages are due to the ultrasensitive spatial registration of EV-bound magnetic nanolabels over the entire volume of the test strip. The tool is promising not only for EV quantification but can be extended for on-site liquid-biopsy analyses of other tumor-related biomarkers (including in combination with hybrid nanoparticles [48,49]) in routine cancer screening and estimation of therapy response.

## 2. Materials and Methods

### 2.1. Cell Culture

Human colorectal adenocarcinoma cell line HT29 (ATCC^®^ HTB-38™; LabMir LTd., Moscow, Russia) and human triple-negative breast cancer cell line MDA-MB-231 (ATCC^®^ HTB-26™; Manassas, VA, USA) were maintained in Dulbecco’s Modified Eagle Medium (DMEM; Sigma, St. Louis, MO, USA) and DMEM-F12 (PanEco, Moscow, Russia), respectively, both supplemented with 10% fetal bovine serum (FBS; Gibco, NY, USA), 2 mM glutamine, and penicillin/streptomycin (1%) (Gibco, NY, USA), at 37 °C in a humidified atmosphere containing 5% CO_2_. For EV production, the MDA-MB-231 and HT29 cells were cultured in 175 cm^2^ and 225 cm^2^ flasks, respectively, until they reached 80–90% confluence; the cells were washed three times with 1 x phosphate-buffered saline (PBS; Sigma, St. Louis, MO, USA) and maintained in 35 mL FBS-free, phenol-red-free DMEM-F12 (or DMEM) for 24 h.

### 2.2. Patient Samples

Ascitic fluids from ovarian cancer patients (clinical stage III) before neoadjuvant chemotherapy and after it (not earlier than 21 days after a routine course) were obtained from the Russian Scientific Center of Roentgenoradiology (Moscow, Russia), Ministry of Healthcare of the Russian Federation. The study was approved by the Ethics Committee of the Russian Scientific Center of Roentgenoradiology (agreement and protocol no. 30-2018/E of 13 November 2018), and written informed consent was obtained from all the patients who participated. Blood samples from breast cancer patients were obtained from the Department of Obstetrics and Gynecology of the University Medical Centre, Freiburg, Germany. The investigation was approved by the Institutional Ethical Review Board of the University of Freiburg (protocol no. 36/12). Written consent was given by all patients. Blood collected in vacutainers was centrifuged at 2500× *g* for 20 min to obtain serum. The serum samples were aliquoted and stored at −80 °C until further processing. EV isolation and their characterization by NTA [50], transmission electron microscopy (TEM) [51], ELISA, and flow cytometry [52] are presented in the Appendix A).

### 2.3. Preparation and Fluorescent Labeling of Antibody-Functionalized Magnetic Nanoparticles

In the research, we used commercial superparamagnetic nanoparticles, namely, 203-nm carboxyl-modified (COOH-) polystyrene-encapsulated iron oxide (50% *w*/*w* polymer/iron oxide) “Bio-Estapor Microspheres” (Estapor–Merck Millipore, Darmstad, Germany). The protocol of conjugation of magnetic nanoparticles with antibody and their further fluorescent labeling is described in detail in [32,53]. Briefly, 3 µL of MP were magnetically washed in deionized water and then in 2-morpholinoethanesulfonic acid (MES; Appli-Chem, Darmstad, Germany) buffer (0.1 M, pH 5.0) with occasional sonication to prevent aggregation. After that, the particles were incubated for 20 min in activation buffer: 5 mg *N*-(3-dimethylaminopropyl)-*N*′-ethylcarbodiimide hydrochloride (EDC; Sigma Aldrich, St. Louis, MO, USA) and 2.5 mg *N*-hydroxysulfosuccinimide sodium salt (sulfo-NHS; Sigma Aldrich, St. Louis, MO, USA) in 50 μL of MES buffer (0.1 M, pH 5.0). Then, 30 μL of mouse monoclonal antibody to CD9 (1 mg/mL, clone MEM 61; Exbio Praha, Czech Republic) in PBS buffer (pH 7.4) was added to MP and incubated overnight at +4 °C in a horizontal shaker. To block the unreacted EDC, 10 μL of 10% bovine serum albumin (BSA; Sigma Aldrich, St. Louis, MO, USA) was added followed by 60 min incubation at continuous stirring. Finally, the MP conjugates with anti-CD9 antibody (anti-CD9–MP) were thrice magnetically washed with PBS buffer (pH 7.4) and stored in PBS with 0.1% sodium azide (Sigma Aldrich, St. Louis, MO, USA) at a final particle concentration of 2 mg/mL at +4 °C before use.

As has been reported previously [54], the magnetic nanoparticles selected for the present research, as well as the procedure of their conjugation with antibodies, produce highly stable conjugates which do not agglomerate and have an excellent performance in assays for more than one year after the conjugation.

Anti-CD9-MPs were labelled with fluorescent dyes: fluorescein isothiocyanate (FITC; Sigma Aldrich, St. Louis, MO, USA) or Sulfo-Cyanine5 NHS ester (sulfo-Cy5-NHS; Lumiprobe, Moscow, Russia). Briefly, 60 µg of anti-CD9-MP in 90 μL of 1% BSA-PBS buffer was mixed with 200 µg of FITC in 10 μL of dimethyl sulfoxide (Sigma Aldrich, St. Louis, MO, USA) or 400 µg of sulfo-Cy5-NHS in 10 µL of PBS and incubated for 4 h at room temperature (RT). The resulting anti-CD9-MP labelled by FITC or sulfo-Cy5-NHS (FITC-anti-CD9-MP or Cy5-anti-CD9-MP, respectively) were magnetically washed five times with PBS buffer (pH 7.4) from the non-bound dye (final particle concentration—2 mg/mL).

### 2.4. Imaging Flow Cytometry

To characterize the complexes of EVs with the antibody-functionalized magnetic nanoparticles (EV–anti-CD9-MP), we used imaging flow cytometry.

First, 3 μL of Cy5-anti-CD9-MP (or FITC-anti-CD9-MP) was added to 30 µL of HT29 EV samples (1.5 × 10^9^ EV) in PBS containing 1% BSA and 0.1% Tween-20 (Sigma Aldrich, St. Louis, MO, USA), pH 7.4, and incubated overnight at RT in a rotator. During the incubation, EVs formed complexes with Cy5-anti-CD9-MP (or FITC-anti-CD9-MP). Then, the analyzed samples, which contained the formed immune complexes and also unbound vesicles, were stained with 10 µL of PE-anti-CD81 (clone M38, Exbio Praha, Czech Republic) or APC-anti-CD81 (cat. no. 551112, BD Biosciences, San Jose, CA, USA) antibodies for 2 h at RT in the dark on a rotator. Finally, the samples were magnetically washed thrice with PBS containing 0.1% Tween-20 (pH 7.4; Sigma Aldrich, St. Louis, MO, USA). For control of specificity of vesicle binding to magnetic nanoparticles, we prepared EV-free samples, namely, samples containing Cy5-anti-CD9-MP (or FITC-anti-CD9-MP) mixed with PE-anti-CD81 (or APC-anti-CD81) antibody, as well as a buffer control.

All samples were analyzed on an imaging flow cytometer Amnis ImageStream X Mark II (Luminex Corporation, Austin, TX, USA) at 40× magnification; 488 nm (200 mW) and 642 nm (150 mW) lasers were used for fluorescence excitation, and a 785 nm laser (70 mW) was used for the side scatter measurements. The flow rate was set to “low speed/high sensitivity” mode. Each sample was acquired for 1 min [55]. The compensation was applied using the data from single-stained polystyrene beads (Magsphere Inc., Pasadena, CA, USA). The images were processed using the IDEAS software provided with the cytometer. A gating strategy is shown in the Appendix A. A histogram of the brightfield area values was used to exclude debris and MP aggregates. Using a side scatter intensity histogram, 1 μm speed beads that constantly passed inside the instrument flow stream were excluded. Then, the MPs were gated using a Cy5 intensity histogram. After all gating steps, 99.5% of events from a buffer-only sample were excluded. To plot the final PE intensity histogram, events from the analyzed samples were normalized to 45,000.

When a sample was analyzed using another pair of fluorescent labels (FITC/APC-anti-CD81), after all gating steps, 99.3% of events from the buffer-only sample were excluded. To plot the final APC intensity histogram, events from the analyzed samples were normalized to 25,000.

The histograms of the Cy5-anti-CD9-MP-bound EVs stained with PE-anti-CD81 (or FITC-anti-CD9-MP-bound EVs stained with APC-anti-CD81) were fitted with two Gaussian distributions using the Origin software (OriginLab Corporation, Northampton, MA, USA).

### 2.5. Design of IC Strips

The IC strips were designed to realize a sandwich format of immunochromatographic assay [34,56] with the addition of a negative control line (NCL) formed by an isotype control antibody. Each test strip was composed of an overlapping sample pad (Ahlstrom CytoSep, Helsinki, Finland), nitrocellulose (UniSart CN95, 260 µm thick and 100-µm backing; Sartorius AG, Goettingen, Germany), and absorbent (Ahlstrom CytoSep, Helsinki, Finland) membranes assembled on an adhesive plastic backing sheet (Lohmann, Hebron, KY, USA). To form the NCL, mouse IgG1 isotype control antibody (1 mg/mL, clone MOPC-21; Exbio Praha, Czech Republic) was deposited onto a 40 mm-wide and 300 mm-long nitrocellulose membrane at 15 mm from the membrane front edge at a jetting rate of 1 µL/cm with a Biodot XYZ3060 Dispense Platform (Biodot Inc., Irvine, CA, USA). The test line (TL) was deposited similarly at 25 mm from the membrane front edge by jetting monoclonal antibodies to CD81 (clone M38; Exbio Praha, Czech Republic) or CD326/EpCAM (clone 323/A3; Exbio Praha, Czech Republic). The control line (CL) was formed by jetting antispecies IgG antibodies (cat. no. 7457507, Lampire, Pipersville, PA, USA). The fully assembled card was dried for 4 h at +37 °C. The IC strips were cut 3 mm wide by an automated cutter, the “CM4000 Guillotine Cutter” (Biodot Inc., Irvine, CA, USA).

### 2.6. Magnetic Particle Quantification

The developed tool is based on a highly sensitive readout technique of magnetic particle quantification [42]. MPQ employs a nonlinear magnetization of magnetic nanoparticles subjected to an alternating magnetic field at two frequencies, with the recording of particle response at a combinatorial frequency. The method is insensitive to linear diamagnetic and paramagnetic materials, such as biological fluids and components of the IC strips. In the MPQ device used in this study, the nanomagnetic IC test strips were subjected to a two-component alternating magnetic field: H = H_1_ × cos(2π · f_1_ · t) + H_2_ × cos(2π · f_2_ · t). One of the field components had the frequency f_1_ = 152 Hz and amplitude H_1_ = 150 Oe, while the parameters of the second component were f_2_ = 156 kHz and H_2_ = 50 Oe. The magnetic field amplitudes chosen were high enough to reach the nonlinearity of magnetization curve M(H). That produced a nonlinear response from the superparamagnetic nanoparticles within the test strips to be recorded at combination frequencies, e.g., f = f_2_ ± 2f_1_. The detailed description of the MPQ principle, as well as the layout of key components and parts of the related instrumentation, can be found in [57,58]. Briefly, an MPQ detector has two generators, the signals from which are amplified and fed to two coaxial induction coils. To quantify MP, the IC test strip is placed inside these coils. The output signal is recorded at a combinatorial frequency from one of the coils connected through a lock-in filter to the amplifier. The generation of combinatorial harmonics using independently measured M(H) along with the benefits of using two frequencies for MP counting in various applications was analyzed in detail in [59].

### 2.7. Preparation of Samples and Immunochromatographic Assay Procedure

The samples were prepared by spiking different amounts of HT29 or MDA-MB-231 vesicle stock solution into buffer (PBS, 1% BSA, 0.1% Tween, pH 7.4). The final EV quantities in the samples were 0, 1.5 × 10^7^, 5 × 10^7^, 1.5 × 10^8^, 5 × 10^8^, and 1.5 × 10^9^ of particles per test. Each EV sample (30 μL) prepared as described above was mixed in a plastic tube with 3 μL of anti-CD9-MP and incubated for 2 h or overnight at RT in a rotator. Then, 90 μL of buffer (PBS, 1% BSA, 0.1% Tween, pH 7.4) was added. The sample pad of the IC strip was immersed in the obtained solution, which migrated along the strip toward the absorbent pad. After that, the IC strip was inserted into an MPQ detector for registration of magnetic signals. The quantity of EV–anti-CD9-MP complexes bound at the test line was proportional to EV concentration in the sample.

For detection of EVs isolated from clinical samples (human serum and ascites), the EV samples (10 μL and 25 μL, respectively) were mixed in a plastic tube with 3 μL of anti-CD9-MP and incubated in buffer (PBS, 1% BSA, 0.1% Tween, pH 7.4) overnight at RT in a rotator. The further assay procedure was as described above.

### 2.8. Data Processing

All experiments were performed at least three times. A signal *S* for each strip was calculated as a difference between the specific signal at the TL and the non-specific signal at the NCL. Such a signal calculated for the strips corresponding to zero EV inputs was considered as a background signal *B*. The standard deviation of these background signals was denoted *STD_bgr_* and the mean of the background signals *B_avr_*. For each EV input, the signal-to-noise ratio (SNR) was calculated as:SNR = (*S* − *B_avr_*)/*STD_bgr_*.(1)

The data in the graphs represent the mean values of SNR, and the error bars the standard deviations of SNR for the corresponding number of replicated measurements for each EV input. The histograms were fitted with two Gaussian distributions using Origin software (OriginLab Corporation, Northampton, MA, USA).

The limit of detection (LOD) was determined as the EV quantity, at which SNR equaled 2 [60].

## 3. Results

### 3.1. Principle of the Nanomagnetic IC Tool

The proposed tool realizes the sandwich format of immunochromatographic assay with antibody-functionalized MPs as labels detectable by an original ultrasensitive technique of magnetic particle quantification [42] by their non-linear magnetization (Figure 1). The assay was demonstrated with EVs sandwiched between a tracer antibody (anti-CD9) conjugated with MPs and a capture antibody (anti-CD81 or anti-EpCAM) deposited on the TL (Figure 1B). A negative control line formed by an isotype control antibody was for the first time introduced to the test strip to account for even subtle non-specific contributions to the signal. A positive control line is formed by IgG that recognizes tracer antibodies. A sample containing complexes of EVs with antibody-functionalized MPs (e.g., anti-CD9-MPs) migrates along the test strip under capillary forces and passes sequentially NCL, TL, and CL. Upon antigen–antibody recognition, the complexes are retained by a specific capture antibody at the TL, and the unbound anti-CD9-MPs migrate further to the absorbent pad (Figure 1C). Thus, the magnetic signal at the TL is proportional to the quantity of EV-bound antibody-functionalized MP complexes specifically retained at the TL. The magnetic signals from the IC strip are read out with an MPQ detector (Figure 1D).

In this research, we have applied this tool for quantitative detection of EVs purified from different types of samples, namely, cell culture supernatants, human serum, and ascites.

### 3.2. EV Characterization

First, we used EVs isolated from the cell culture supernatants of human colorectal adenocarcinoma cell line HT29 and breast cancer cell line MDA-MB-231 (the isolation protocol is available in the Appendix A). Small EVs were enriched by ultrafiltration and characterized according to the Minimal Information for Studies of Extracellular Vesicles guidelines [61] using TEM, NTA (Figure 2A–D; NTA and TEM measurements are detailed in the Appendix A), and flow cytometry (the protocol and EV characterization are available in the Appendix A). TEM visualization of EVs revealed characteristic EV-like structures of 60–120 nm in diameter (Figure 2A,C). Quantitative evaluation of these images was consistent with the NTA data (Figure 2B,D).

The NTA and ELISA characterizations of EVs derived from body fluids of cancer patients can be found in the Appendix A.

### 3.3. Visualization of Formation of “EV–Antibody-Functionalized MP” Immune Complexes by Imaging Flow Cytometry

Efficient binding of EVs with antibody-functionalized MPs during the incubation time is essential for proper performance of the proposed nanomagnetic IC tool. This factor was analyzed by imaging flow cytometry. In these experiments, antibody-functionalized MPs (anti-CD9-MPs) were labeled with sulfo-Cy5-NHS or FITC fluorophores and are referred to below as Cy5-anti-CD9-MP and FITC-anti-CD9-MP, respectively. After the incubation, EVs bound to Cy5-anti-CD9-MP (or FITC-anti-CD9-MP) were further immunolabeled with PE-anti-CD81 (or APC-anti-CD81). As controls, we used samples containing Cy5-anti-CD9-MP, Cy5-anti-CD9-MP mixed with PE-anti-CD81, as well as a buffer control. Thus, four types of the samples were analyzed: (1) Cy5-anti-CD9-MP; (2) Cy5-anti-CD9-MP with PE-anti-CD81; (3) Cy5-anti-CD9-MP with EV and PE-anti-CD81; (4) buffer only.

Figure 3 shows APC histograms for the samples 1–3 and representative images of the immune complexes formed by Cy5-anti-CD9-MP with PE-anti-CD81-labelled HT29 EVs. The images for Cy5-anti-CD9-MP and Cy5-anti-CD9-MP with PE-anti-CD81 can be found in the Appendix A. The histogram for the buffer-only sample is not shown because almost all events from this control sample have been excluded as a result of gating.

Based on the imaging data, we quantified the fraction of anti-CD9-MPs that formed the immune complexes with EVs. As follows from the fitting data, 38% of the population of Cy5-positive events were also PE-positive. Similar results were obtained for the APC-anti-CD81-stained EVs and FITC-anti-CD9-MPs (Appendix A). In the latter case, 30% of the population of FITC-positive objects were also positive for APC. The data indicate that the EV–anti-CD9-MP interactions during the incubation time were efficient and highly specific.

### 3.4. Analytical Performance of the Developed Nanomagnetic IC Tool for EV Quantification

The proposed tool is demonstrated here by quantitative EV detection using CD9, CD81, and EpCAM membrane surface proteins as targets. The photographs of the IC strips and the distribution of MPQ signals along the test strips at different EV inputs are shown in Figure 4.

Figure 4 indicates that the traditional optical readings of MPs by TL coloration are less efficient than the MPQ readouts. While the colored line is hardly visible in the IC strips at 15 × 10^7^ EV/test, MPQ features a strong signal at the TL position in the MP distribution even with an input one order of magnitude less of 1.5 × 10^7^ EV/test.

That is due to much higher sensitivity of MPQ and its ability to count MPs within the entire volume of TL of the nitrocellulose membrane rather than merely from its surface at the optical readout. The magnetic signal of the proposed IC tool was calibrated using EVs isolated from the HT29 and MDA-MB-231 cell culture supernatants. EV samples with inputs of 0, 1.5 × 10^7^, 5 × 10^7^, 1.5 × 10^8^, 5 × 10^8^, and 1.5 × 10^9^ of particles per test were used. The detection limits for CD81^+^/CD9^+^ HT29 and MDA-MB-231 EVs calculated from the calibration plots (Figure 5) are 1.1 × 10^7^ and 1.3 × 10^7^ particles/test (3.7 × 10^5^ and 4.3 × 10^5^ EV/µL), respectively. The dynamic range exceeds two orders of magnitude. The LODs are one to two orders of magnitude better than those achieved by the most sensitive LF systems [40,41] reported so far.

The calibration plots for EpCAM^+^/CD9^+^ EVs isolated form the HT29 and MDA-MB-231 cell culture supernatants are shown in the Appendix A.

Further, we studied the influence of duration of EV incubation with Anti-CD9-MPs on the tool’s sensitivity (for HT29 EV as an example, see Figure 6). It was found that at 2 h incubation, LOD was 2.4 × 10^7^ for CD81^+^/CD9^+^ EVs per test. As follows from Figure 6A, reliable detection of 2.4 × 10^7^ CD81^+^/CD9^+^ HT29 EVs/test does not require overnight incubation. However, as expected, at a low vesicle input (1.5 × 10^7^), the overnight incubation improves the assay sensitivity (Figure 6B).

### 3.5. Quantification of EVs Isolated from Body Fluids of Cancer Patients

The developed nanomagnetic IC tool was used for quantification of EVs isolated from ascites fluid of patients with ovarian cancer (two samples, A1–A2), from human serum of patients with breast cancer (three samples, B1–B3), and a healthy donor (sample H1). To find the quantity of CD81^+^/CD9^+^ EVs purified from the clinical samples, we matched the obtained values of magnetic signals to the calibration plot for CD81^+^/CD9^+^ MDA-MB-231 vesicles (Figure 7).

## 4. Discussion

The great potential of extracellular vesicles as diagnostic and prognostic biomarkers for liquid biopsy [3,8,9,12] and the limitations of available analytical methods for EV quantification call for novel techniques, which would be attractive for routine clinical registration of EVs derived from complex bodily fluids.

A simple-to-use and cost-efficient tool has been developed that employs an advantageous combination of ultrasensitive MPQ registration with a novel design of immunochromatographic test strips for highly sensitive EV quantification using small-volume samples with high specificity in a wide dynamic range. We have demonstrated the tool with EVs isolated from various mediums, including cell culture supernatants and biofluids of patients with breast (serum) and ovarian (ascites) cancers.

Our tool offers two orders better sensitivity than conventional ELISA, which is commonly used for EV quantification [14,15]. The fundamental factors limiting ELISA sensitivity for EVs isolated from body fluids are slow EV diffusion and high nonspecific adsorption of biomolecules from the complex mediums. The commercial ELISA kits offer sensitivity on the level of 10^9^ EVs/test. Furthermore, the user-friendly procedure of our tool is shorter and does not involve multiple steps and washing procedures. Importantly, the duration of EV incubation with magnetic nanolabels has no significant effect on the tool’s sensitivity (Figure 6), e.g., 2 h incubation allows reliable detection of 2.4 × 10^7^ CD81^+^/CD9^+^ HT29 EVs/test. To reach the high sensitivity, our tool does not need integration with other analytical techniques and/or devices for signal amplification, unlike the recently proposed ELISA-based methods [62,63,64], which involve microfluidic and microchip technologies and entail further optimization for liquid biopsy.

The MPQ detector registers the EV-bound magnetic nanolabels within the entire volume of the test strip rather than only on its surface as under optical readings applied in other LF methods of EV detection. As a result, we achieved a 30-fold better sensitivity with respect to the recently reported LF immunoassay based on magnetic nanoparticles registered optically [40] or by impedance measurements [41], as well as a one-order improvement with respect to the LF assay based on gold nanolabels [40]. Furthermore, our test strips do not undergo test line discoloration and can be read at any time rather than within 10–15 min after the test as with optical readings of gold nanoparticles. This is an attractive feature for testing in remote regions.

We show that the recorded magnetic signal depends exclusively on the quantity of EVs rather than on probable variations in the number of membrane antigens on their surfaces. Indeed, the measured SNR dependences exhibited very similar values and LODs for the same EV inputs (Figure 5 for CD81^+^/CD9^+^ HT29 and CD81^+^/CD9^+^ MDA-MB-231 EVs; Appendix A for EpCAM^+^/CD9^+^ HT29 and EpCAM^+^/CD9^+^ MDA-MB-231 EVs). These results agreed well with the flow cytometry results for the expression of the membrane antigens on HT29 and MDA-MB-231 cells (Appendix A for CD9 and CD81; Appendix A for EpCAM). The flow cytometry data indicate that the cells of both types were totally positive for CD9 and CD81 markers and that 100% of HT29 cells and 95% of MDA-MB-231 cells were EpCAM-positive. Notably, the observed average value of PE fluorescence for EpCAM was lower with MDA-MB-231 cells compared to HT29 cells. That may have been due to a smaller quantity of EpCAM antigens on each MDA-MB-231 cell. The slightly better LOD for EpCAM^+^/CD9^+^ MDA-MB-231 EVs matches this assumption. For immunochromatographic assays, the lower amounts of membrane antigens used as targets may be beneficial. Their high density on EV surfaces may facilitate aggregations during immunochemical reactions and sterically impede EV recognition by the capture antibody at the TL. These issues may be especially pronounced when using abundant EV antigens as targets for the tracer antibody (conjugated with the label) and scarce EV antigens for recognition by capture antibodies deposited at the TL [65]. Meanwhile, steric hindrance will not contribute significantly to further binding of the “label–tracer antibody–EV” immune complexes with the capture antibody if abundant protein markers (e.g., tetraspanins) are used for interactions with both tracers and capture antibodies. Due to the high availability of tetraspanin epitopes on EVs, if some epitopes are sterically shielded, the others will be accessible for immune recognition. Here, we demonstrate that anti-CD9/anti-CD81 employed as tracer/capture antibodies produce higher sensitivity than that achieved by other LF systems which use different pairs of tetraspanin-specific antibodies (anti-CD63/anti-CD9 or anti-CD63/anti-CD81) or an opposite combination (anti-CD81/anti-CD9) [40,65]. Besides the selection of antibody combination, other assay parameters, e.g., the number of antibodies for conjugation with nanoparticles, the number of conjugates per test, the duration of EV incubation with the conjugates, etc., should also be optimized.

Additional factors that may affect the tool’s sensitivity are the efficiency and specificity of EV binding with the functionalized magnetic nanolabels during incubation. These parameters of the “EV–anti-CD9 MP” bonds were validated by high-resolution imaging flow cytometry. This technique is widely used for analysis of small objects (<200 nm), including EVs or various fluorescent beads, to confirm particle morphology with unambiguous differentiation from aggregates and cell debris, and permits analysis of cell/particle interactions [27,28,55,66,67]. The IFC data also verified the optimality of the used parameters (the number of magnetic particle conjugates, incubation time) for the efficient IC assay.

The diagnostic value of liquid biopsy is in the quantification of EVs isolated mainly from complex biofluids (serum, plasma, ascites, etc.) of cancer patients. According to the SNR dependences for HT29 и MDA-MB-231 EVs, a remarkable feature of our tool is a possibility to quantify EVs, derived from physiological fluids, using the calibration plot obtained for EVs isolated from cell culture supernatants. It is known that the accurate quantification of EVs isolated from serum and other body fluids of patients and healthy donors is hampered by high contents of interfering components such as lipoproteins and protein aggregates. Our tool can perceive even subtle non-specific contributions to the signal at every measurement through an isotype control line introduced for the first time to the test strip for EV determination. Such negative control is accurate as the MPQ detectors record the magnetic signal distribution along the test strip during a short single-pass readout of the strip (Figure 4B). Here, we have shown that the developed tool permits accurate quantification of EVs isolated from complex biological fluids of cancer patients (Figure 7).

Despite technological advances, highly sensitive analytical techniques for EV quantification and characterization (SPR [21,22], SERS [21,24,25], fluorescent NTA [26], etc.) are still labor- and time-consuming, requiring sophisticated, expensive instruments and highly qualified operators. The tool presented here does not need any of these. The MPQ detectors are portable, so the tool can be further adapted for out-of-laboratory or field conditions for point-of-care liquid biopsy, especially for non-invasively obtained biofluids [68,69].

## 5. Conclusions

In the present study, we have developed a nanomagnetic IC tool for quantitative detection of extracellular vesicles. The major benefits of the tool include: an attractive limit of detection of 3.7 × 10^5^ particles/µL, small-volume samples, high sensitivity (slope of calibration curve) and specificity for determination of CD81^+^/CD9^+^ and EpCAM^+^/CD9^+^ EVs derived from HT29 and MDA-MB-231 cell culture supernatants, and accurate quantification of EVs isolated from complex biological fluids of cancer patients (serum, ascites). The tool is easy-to-use, cost-efficient, and can be implemented at point of care. It is promising for highly sensitive quantification of EVs for early diagnostics, screening for cancer, and can be employed for routine measurement of tumor-related biomarkers of different types for liquid biopsy.

The tool’s further evolution may involve developing multi-parameter formats for simultaneous assessment of various disease-associated EV surface markers to estimate their relative expression levels along with highly sensitive quantification of populations of specific EVs. Other possible improvements include reducing the assay time down to 30 min, further enhancing the tool’s sensitivity, and extending the dynamic range. These advances can be realized via efficient combinations of tracer/capture antibodies or aptamers as well as improved magnetic nanolabels to be registered by upgraded MPQ detectors. In addition, the proposed tool can be extended for detection of other biomarkers (proteins, nucleic acids) by using respective specific recognition biomolecules.

## Figures and Tables

**Figure 1 nanomaterials-12-01579-f001:**
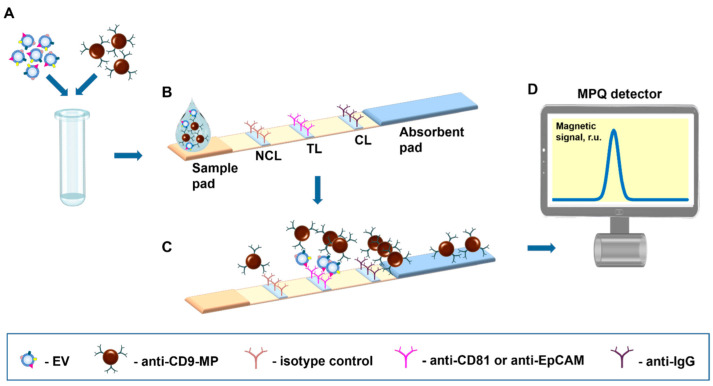
Scheme of the nanomagnetic IC tool for EV quantification. (**A**) Samples are incubated with magnetic nanoparticles functionalized with an EV-specific antibody. (**B**) Test strip design. (**C**) Immunochemical reactions during the migration along the test strip of the sample containing EVs–anti-CD9-MP complexes. (**D**) The test strip is inserted into an MPQ detector to register magnetic signals from the nanolabels.

**Figure 2 nanomaterials-12-01579-f002:**
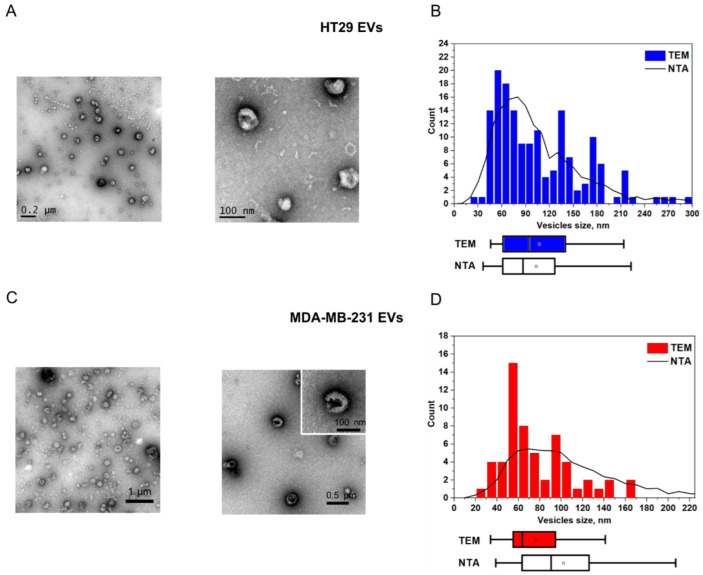
Characterization of EVs purified from HT29 and MDA MB-231 cell culture supernatants. (**A**,**C**) TEM images and (**B**,**D**) corresponding NTA/TEM particle size distributions. The squares indicate mean values, while the box-and-whisker plots show 5%, 25%, 50%, 75%, and 95% percentiles.

**Figure 3 nanomaterials-12-01579-f003:**
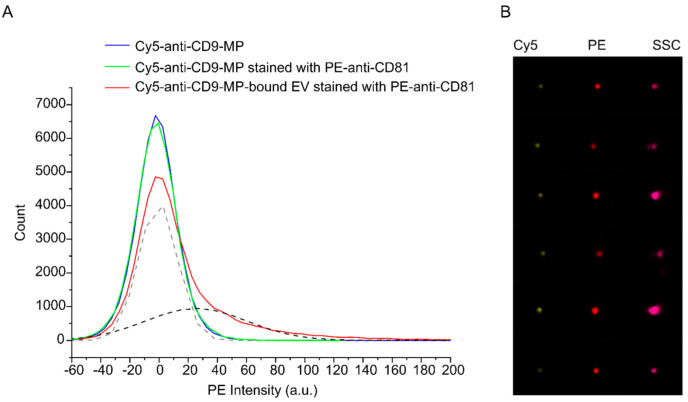
Analysis of binding of PE-anti-CD81-stained HT29 EVs to Cy5-anti-CD9-MP with imaging flow cytometry. (**A**) PE intensity histograms. Fitting of the histogram of the Cy5-anti-CD9-MP-bound EVs stained with PE-anti-CD81 with two Gaussian distributions are shown by dashed lines (grey—unbound Cy5-Anti-CD9-MP; black—EV–Cy5-Anti-CD9-MP immune complexes). (**B**) Representative images of the formed immune complexes between Cy5-anti-CD9-MP and PE-anti-CD81-labelled EVs in Cy5, PE, and side scatter (SSC) channels.

**Figure 4 nanomaterials-12-01579-f004:**
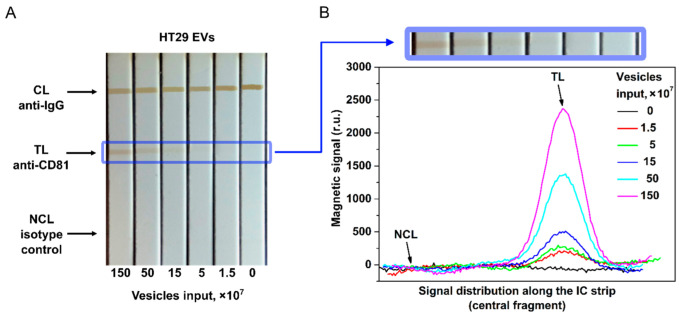
EV quantification by the developed nanomagnetic IC tool. (**A**) Photograph of the IC strips for HT29 EVs with anti-CD81 as capture antibody and anti-CD9 as tracer antibody. (**B**) Signal distribution along the IC strip at different EV inputs (data for CD81^+^/CD9^+^ HT29 EVs).

**Figure 5 nanomaterials-12-01579-f005:**
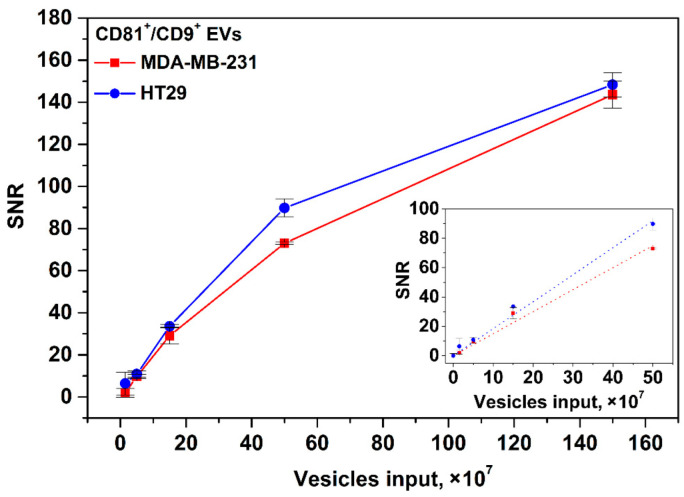
Calibration plots for CD81^+^/CD9^+^ HT29 and MDA-MB-231 EVs. Insert: zoomed linear fitting plots for low EV input range.

**Figure 6 nanomaterials-12-01579-f006:**
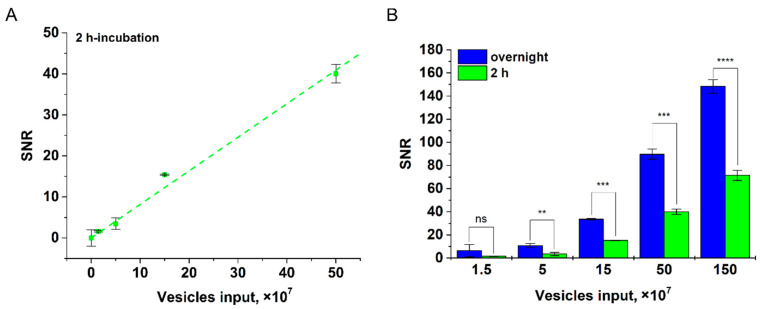
Detection of CD81^+^/CD9^+^ HT29 EVs at various incubation times. (**A**) Linear fitting plot at 2 h incubation. (**B**) Dependences of SNR upon incubation time (2 h and overnight) with different HT29 EV inputs; statistical significance determined using the unpaired two-tailed Student’s *t*-test is denoted by asterisks (ns—*p* > 0.05, **—*p* < 0.01, ***—*p* < 0.001, ****—*p* < 0.0001).

**Figure 7 nanomaterials-12-01579-f007:**
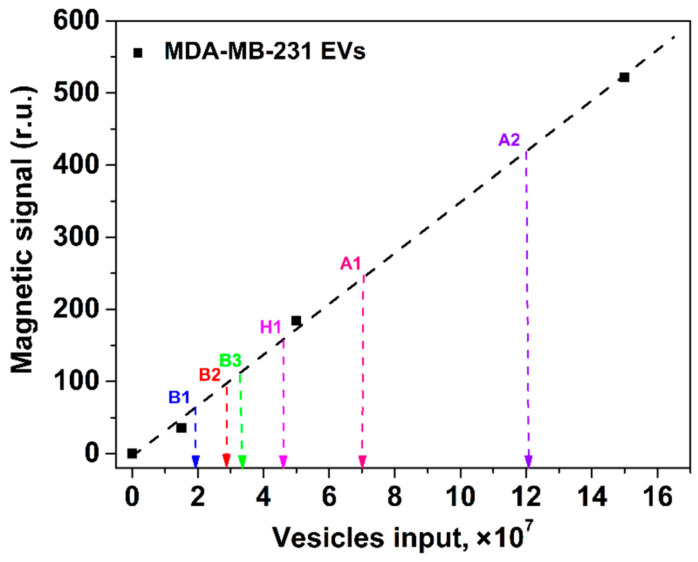
Quantification of EVs isolated from clinical samples with the proposed nanomagnetic IC tool using the calibration plot (shown by the black dashed line) for CD81^+^/CD9^+^ MDA-MB-231 EVs. Clinical samples: serum from patients with breast cancer (B1–B3), a healthy donor (H1), and ascites fluids of patients with ovarian cancer (A1, A2).

## Data Availability

The data presented in this study are available on request from the corresponding author.

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
