# Peer review of "Highly Sensitive Nanomagnetic Quantification of Extracellular Vesicles by Immunochromatographic Strips: A Tool for Liquid Biopsy"

_nanomaterials, 2022, doi:10.3390/nano12091579_

Round 1

Reviewer 1 Report

EVs share some general membrane surface proteins that are potential for non-invasive diagnosis of tumors, evaluation of therapy response and the active targeting of nanodrugs. Development of ultrasensitive temporal-spatial analysis methods for these biomarkers in EVs provides a more accurate quantification of the total EV population for liquid biopsy. Recently, the nanoparticles, particularly nanomagnetism and nanoplasmonics, labeling IC tools have shown promising for on-site liquid-biopsy analyses in routine cancer screening and estimation of therapy response. In this article, authors present a highly-sensitive and easy-to-use nanomagnetic immunochromato- graphic (IC) tool for quantification of extracellular vesicles (EVs), which show great potential as liquid biopsy agents, with a detection limit of 3.7 × 105 EV/µL, better than the most sensitive traditional lateral flow system and commercial ELISA kits. Due to the spatial quantification of EV-bound magnetic nanolabels, the IC strips contain an isotype control to ensure detection specificity and can be read out any time after test. The article is generally written well and their developed nanomagnetic labeling IC strips shall have great promising for liquid biopsy in daily clinical routine for the related biomarker detection. Before publication, there are some suggestions for authors to consider to improve their articles for more readable and attractive.

  • What kind of magnetic nanoparticles are they using, composition? Please mention here and give a brief synthesis process for readers’ convenience, besides citing their articles (36, 51). Better to provide one TEM image in this article to show the dispersion status. Please referring to the article:Xiaoxiong Zhao, et al., Nanoscale Advances, 2022, 4, 190-199ï¼›
  • Usually, the magnetic nanoprobes are prone to assemble or aggregate during application or storage, particularly for those so large size (203-nm). How do they dealt with this problem in the preparation of their IC strips, making them uniform dispersion, which is very important for the detection sensitivity and accuracy.
  • The conjugation reactions for the IC strips fabrication are important for readers to understand the key technique. Please detailed reaction by referring to the articles: Weiwei Zhang, et al., Chem. Mater. 2020, 32, 5044-5056.
  • The highly sensitive readout technique is key for this IC tool. Please give a layout of this instrumentation with key components and parts and briefly describe the key detection mechanism, not only offering the references: 43, 46 and 53-57. It is important to improve the readable and attractive for more readers.
  • It is better to show the real image of the IC stripes and then the magnified key details, in Figure 1 and/or Figure 2, or Figure 4.

Reviewer 2 Report

This manuscript (Bragina, V et al) is reporting magnetic nanoparticles engineered as a targeting reagent in immunochromatography for biomedical applications. It has demonstrated its potential utility in the quantification of extracellular vesicles prepared from a few cancer cells. It has showed strong impactful results through its design features, convenience for use and detection performance such as high sensitivity increased by one to two orders of magnitude compared to existing methods.

One minor comment is to discuss its performance or perspective on testing EVs of non-target cells such as those from normal healthy cells.

Overall, this manuscript is written clearly and reads well. It is strongly recommended for publication in nanomaterials.

Reviewer 3 Report

Braguin et al. presented immunochromatographic strips for the detection of extracellular vesicles with magnetic particle quantification (MPQ) systems. Compared to conventional detection methods, they achieved an improvement of 1 to 2 orders of magnitude. However, the MS need to be revised for its clarity. In addition, English needs to be thoroughly revised to deliver the strength of their data. Therefore, major revisions are needed to improve the manuscript.

Major comments

  • The standard deviation and significance of each data set were not included. Therefore, the authors must modify the figures (e.g. figured 2, 6).
  • As shown in the schematic (figure 1), there are three detection zones (NCL, TL and CL). However, the control line (CL) does not appear in the results shown in figure 4A. The author should include the control line.
  • The abstract was poorly written, and unnecessary words such as "topical application" were used. What is the relationship between topical application and detection of extracellular vesicles? The abstract should provide background, issues, research objectives, results obtained, and conclusions in chronological order.
  • The logical arrangement of the introduction needs further improvement. A proper and detailed description of the MPQ system is important for a better understanding of the manuscript. Therefore, the author needs to add a description of the MPQ system.

Minor comments

  • Repeated word “medium” line 104 and 110.
  • The authors need to include city and state of the company (e.g., Sigma Aldrich, Louis, MO, USA).
  • Section 2.6 should show the method used for quantifying the magnetic nanoparticles. Remove line 214-224, it is an introduction about MPQ.
  • The manuscript lacks conclusion, drawback, and future prospect.

Round 2

Reviewer 3 Report

The revised MS include all the necessary parts for the publication. I think it is ready for the publication.